# On the meaning of independence in climate science

James Annan and Julia Hargreaves

BlueSkiesResearch.org.uk
Settle, UK

*Correspondence to:* James Annan (jdannan@blueskiesresearch.org.uk)

**Abstract.**

The concept of independence has been frequently mentioned in climate science discussions, but has rarely been defined and discussed in a theoretically robust and quantifiable manner. In this paper we argue that any discussion must start from a clear and unambiguous definition of what independence means and how it can be determined. We introduce an approach based on the statistical definition of independence, and illustrate with simple examples how it can be applied to practical questions. Firstly, we apply these ideas to climate models, which are frequently argued to not be independent of each other, raising questions as to the robustness of results from multi-model ensembles. We explore the dependence between models in a multi-model ensemble, and suggest a possible way forward for future weighting strategies. Secondly we discuss the issue of independence in relation to the synthesis of multiple observationally-based constraints on the climate system, using equilibrium climate sensitivity as an example. We show that the same statistical theory applies to this problem, and illustrate this with a test case, indicating how researchers may estimate dependence between multiple constraints.

## 1   Introduction

Approximately 30 climate models contributed to recent iterations of the CMIP databases, and they generally agree, at least on broad statements: the world is warming, anthropogenic emissions of $CO_2$ is the major cause of this, and if we continue to emit it in large quantities then the world will continue to warm at a substantial rate for the foreseeable future (Stocker et al., 2013). The consensus across models is also strong for more detailed statements regarding, for example, the warming rates of land versus ocean, high versus low latitudes, and the likely changes in precipitation over many areas. Even where models disagree qualitatively amongst themselves (for example, concerning changes in ocean circulation and some regional details of precipitation patterns), their range of results is still quantitatively limited. Climate models are probably the most widely-used tool for predicting future climate changes, and their spread of results is commonly used as an indication of what future changes might occur.

But should this consensus between models really lead to confidence in these results? If we were to re-run the same scenario with the same model 30 times, we would get the same answer 30 times, whether it be a good or bad model. This repetition of one experiment would not tell us how good the model is, and the behaviour of the real climate system would almost certainly lie outside this narrow range of results. Different model development teams share code, and even if the code is rewritten from scratch, the underlying algorithms and methods are often linked (Knutti et al., 2013). Furthermore, many fundamental theories

are common across all models. So how much confidence can we draw from the fact that multiple models provide consistent answers? How likely is it that common biases across all models are greater than their spread of results, such that the ensemble range does not provide trustworthy bounds on the behaviour of the climate system? These questions have proved difficult to answer, and indeed there appears little consensus as to how we can even address them. Further related issues arise from the increasingly prevalent situation where a single modelling centre contributes multiple simulations to the CMIP archive, some of which may only differ in terms of the settings of uncertain parameters in the climate model, or even just the initial state of the atmosphere/ocean system. A common heuristic when performing multi-model analyses based on a generation of the CMIP ensemble has been to use a single simulation from each modelling centre (e.g. Leduc et al., 2016), but it is not clear where to draw the line when different centres may have shared a common core or sub-models. Is there a better way to select models, and should we use a weighted ensemble? In this case, further questions arise as to how the weights should be defined, in terms of either model performance relative to observations of the real climate, or else in terms of their relationship to other models, or some combination of both. Another related question that has been posed in recent years, is whether the scientific community could collaboratively design or select ensemble members to contribute to CMIP in a more rational and scientifically defensible way than the current ad-hoc 'ensemble of opportunity'. It may be possible to address this issue in terms of statistical sampling and experimental design, but appropriate methods and even language do not yet appear to be well developed in this area.

In Part I of this paper, we consider this question of model independence and discuss how it may be addressed in a mathematically precise and well-founded manner. We present an approach which links the usage in climate science to the statistical definition of independence. We start by reviewing, in Section 2, how the concept of independence has been discussed in the recent literature. In Section 3 we present a theoretical and statistical viewpoint of independence within the Bayesian paradigm, which we argue has direct relevance to this question. We consider how this statistical viewpoint relates to the question of model independence in Section 4, and also present some ideas for how to make practical use of these ideas. We emphasise, however, that the purpose of our paper is to provide a direction and motivation for future investigations rather than attempting to present a complete solution.

In Part II, we illustrate how the theoretical basis for statistical independence can also apply to the question of synthesising observational constraints on the behaviour of the climate system, particularly the equilibrium climate sensitivity. The equilibrium climate sensitivity $S$ represents the equilibrium change in global mean surface temperature following a doubling of atmospheric $CO_2$, and while this parameter is far from a comprehensive description of our future climate, it is commonly used as a summary of the potential magnitude of changes which we might observe in the long term. Different approaches have been proposed for constraining $S$, for example using data drawn from the modern instrumental period, or looking to the paleoclimate record and particularly the Last Glacial Maximum (19–23ka) where global temperatures were far below those of the present day for a sustained period, or searching for constraints that emerge when process studies examine how well different models simulate various aspects of the climate system such as seasonal and interannual variation. It has previously been proposed that multiple constraints can be considered 'independent' and the resulting constraints combined into an overall estimate (Annan and Hargreaves, 2006). However, the principles underlying this approach have not be clearly investigated. In Section 6 we con-

sider how this problem has been addressed in the previous literature, and in Section 7 we consider how the statistical principles apply in both theoretical and practical terms by means of a simple example.

# Part I

# Climate model independence

 ## 2   The literature on model independence in climate research

The question of independence has featured widely in climate research, but the research community has not yet arrived at a clear and unambiguous definition. Different authors have approached the question of independence in different ways, and their approaches are often mutually inconsistent.

One common approach has been to interpret model independence as meaning that the models can be considered as having errors which are independent, identically distributed (i.i.d. in common statistical parlance) samples drawn from some distribution (typically Gaussian) with zero mean (Tebaldi and Knutti, 2007). This is the so-called 'truth-centred' or 'truth plus error' hypothesis. Although it has not generally been explicitly stated, even a small ensemble of samples drawn from such a distribution would be an incredibly powerful tool. If we could sample models from such a distribution, then we could generate arbitrarily precise statements about the climate, including future climate changes, merely by proceeding with the model-building process indefinitely and taking the ensemble mean. This would obviate the need both for computational advances and also for any additional understanding of how to best simulate the climate system. As an illustration of the power of such a (hypothetical) truth-centred ensemble, if the 19 CMIP3 models listed in Table 8.2 of Randall et al. (2007) provided independent (in this sense) estimates of the equilibrium climate sensitivity $S$, then we could immediately generate a 95% confidence interval for the real value for $S$ of $3.2 \pm 0.3°$C based on the assumption that the samples are drawn from a Gaussian distribution of *a priori* unknown variance.

However, the truth-centred hypothesis is clearly refuted by numerous analyses of the ensemble. In particular, the errors of different models are observed to be strongly related, as can be shown by the positive correlations between spatial patterns of biases in climatology (Knutti et al., 2010, Figure 3). As a corollary of this, although the mean of the ensemble generally outperforms most if not all the ensemble's constituent models (Annan and Hargreaves, 2011b), it does not actually converge to reality as the ensemble size grows. Rather, the ensemble mean itself appears to have a persistent and significant bias. There have been some attempts to compensate for this shared bias, for example by estimating the number of 'effectively independent' models contained in the full ensemble (Jun et al., 2008a, b; Pennell and Reichler, 2010). However, the theoretical basis for these calculations does not appear to be clearly justified, and the results presented would, if valid, have startling implications. For example, if we accept the arguments of Pennell and Reichler (2010) that the CMIP3 ensemble contains 8 'effectively independent' models then its full range of sensitivity values, 2.1–4.4°C, would still be a legitimate 99% confidence interval for the true sensitivity, as the probability of 8 independent (in this sense) estimates all simultaneously falling either below or above

the truth is only 1 part in $2^7$. The same argument would apply to any other output or derived parameter of the model climates. That is, we could be 'virtually certain' (to use the IPCC calibrated language) that the model ensemble bounds multiple aspects of the behaviour of the climate system, even with this very modest number of number of 'effectively independent' models. This confident conclusion does not seem very realistic when we consider the limitations which are common to all climate models,

and therefore we are forced to question the appropriateness and validity of the assumptions underlying such analyses.

Abramowitz and Gupta (2008) define independence purely in terms of inter-model differences and suggest down weighting models that are too similar in outputs. This approach has the potential weakness that models that agree *because they are all accurate* will be discounted, relative to much worse models, without any allowance being made for their good performance relative to reality. A challenge for this and related approaches is that the use of a distance measure does not readily suggest

a threshold at which models can be considered *absolutely* independent. All models are designed to simulate the real climate system, and are tuned towards observations of it (Hourdin et al., 2016). Therefore it should not be surprising that climate models appear broadly similar, since the maximum distance (in any relevant metric space) between a pair of models can be no more than the sum of the distances between each of these models and reality. Bishop and Abramowitz (2013) use the pairwise correlations of model errors in their analysis, but only after first debiasing the model simulations, and thus exclude *a priori* one

of the factors which is usually considered a fundamental aspect of both model performance and model similarity.

Some approaches to model independence have been less quantitative in nature. Masson and Knutti (2011) define their interpretation as "independent in the sense that every model contributes additional information" but information in this context is not further defined or quantified. In fact the cluster analysis presented by Masson and Knutti (2011) may be more precisely described by the phrasing in the related paper by Knutti et al. (2013), which states that independence is used "loosely to

20 express that the similarity between models sharing code is far greater than between those that do not". While this pair of papers certainly establishes that point convincingly, there is again no indication of how much similarity should be expected or tolerated between truly 'independent' models, or whether absolute independence is even a meaningful concept in their terms. The interesting philosophical discussions of Parker (2011) and Lloyd (2015) both consider the interpretation and implications of consensus across an ensemble of models that are not independent, but the premise of model dependence is adopted from the

25 literature and these two authors do not themselves attempt to further define this term in a quantifiable manner.

Perhaps the most constructive and complete approach to date is that of Sanderson et al. (2015). In this work, dependence is again defined in terms of inter-model differences in output, and this distance measure is used to remove or downweight the models which are most similar to other models in output. By comparing the inter-model distances both to model-data differences, and to what might be expected by chance with independent samples from a Gaussian distribution that summarises

the full distribution, the authors introduce a threshold at which they argue model differences may be considered appropriately large. However, the epistemic nature of their resulting ensemble is unclear and the resulting reduced ensemble is still only described in terms of *reducing* rather than eliminating dependency.

To summarise, the literature presents a strong consensus that the models are not independent, but does not appear to present such a clear viewpoint concerning what to do about this, or even the precise meaning of this term. Given this lack of clarity,

it is perhaps unsurprising that the IPCC does not address this topic in detail, while nevertheless acknowledging its importance (Cubasch et al., 2013, 1.4.2). Thus, we see not only the opportunity, but also the necessity, of making further progress.

## 3 The statistical context for independence

In probability theory, independence has a straightforward definition. Two events $A$ and $B$ are defined to be independent if the probability of $A$, $P(A)$, is not affected by the occurrence of $B$, so that $P(A|B) = P(A)$ (e.g. Wilks, 1995, Section 2.4.3). Since the joint probability of both events $P(A \cap B)$ is given by $P(A|B)P(B)$, we see that two events are independent if their joint probability is equal to the product of their individual probabilities, i.e. if $P(A \cap B) = P(A)P(B)$. Independence is therefore a symmetric property: $A$ is independent of $B$ if and only if $B$ is independent of $A$. The concept of independence can also be generalised to the case of conditional independence: two events $A$ and $B$ are defined to be conditionally independent given a third event $S$, if their joint probability conditional on $S$, $P(A \cap B|S)$, is equal to the product of their individual probabilities both conditional on $S$, $P(A|S)P(B|S)$. Independence and conditional independence generalise naturally both to continuous distributions $p()$, which is more appropriate for the situations considered in this paper, and also to more than two events.

As we have seen in Section 2, much research on model independence either ignores or explicitly disavows any direct link to this mathematical/statistical definition. Conversely, the primary goal of this manuscript is to argue that this definition must be central to any usable, quantitative theory.

Bayes' Theorem tells us how to update a prior probabilistic estimate of an unknown, $p(S)$, in light of some observations or event $A$ via the equation

$$p(S|A) = p(A|S)p(S)/p(A). \tag{1}$$

$p(A|S)$ is known as the likelihood function (particularly when $A$ is held fixed, and $S$ treated as a variable).

If we have two events $A$ and $B$ then the corresponding equation is

$$p(S|A \cap B) = p(A \cap B|S)p(S)/p(A \cap B). \tag{2}$$

The first term on the right hand side of this equation can be expanded by the laws of probability, resulting in the equivalent formulation

$$p(S|A \cap B) = p(A|B \cap S)p(B|S)p(S)/p(A \cap B). \tag{3}$$

Either of these two equations can in principle be used to calculate the posterior probability of $S$ conditional on both of the events $A$ and $B$, though in practice it may not be straightforward to determine the terms on the right hand sides.

If $A$ and $B$ are conditionally independent given $S$, then $p(A \cap B|S)$ can also be decomposed as $p(A|S)p(B|S)$. Thus in this case,

$$p(S|A \cap B) = p(A|S)p(B|S)p(S)/p(A \cap B). \tag{4}$$

In practice, the term 'independent' is frequently used to refer to conditional independence, especially when $A$ and $B$ are being discussed primarily as observations of, or evidence concerning, some unknown $S$. The significance of this conditional independence is that if we already have likelihoods $p(A|S)$ and $p(B|S)$, then conditional independence allows us to directly create the joint likelihood $p(A \cap B|S)$ by multiplication, rather than requiring the construction of $p(A|B \cap S)$ as an additional step. Inspection of Equations 3 and 4 shows that the conditional independence of $A$ and $B$ given $S$ is equivalent to the condition that $p(A|B \cap S) = p(A|S)$. This equation states that the predictive probability of $A$, given both $S$ and $B$, is equal to the predictive probability of $A$ given $S$. In other words, if we know $S$, then additionally learning $B$ does not change our prediction of $A$. This formulation can be a useful aid to understanding when independence does and does not occur.

## 3.1 The Bayesian perspective

The above elementary probability theory applies equally to the frequentist and Bayesian paradigms. Within the frequentist paradigm, the probability of an event is defined as the limit of its relative frequency over a large number of repeated but random trials. Within the Bayesian paradigm, the probability calculus may be used to describe the subjective beliefs of the researcher. In the remainder of this manuscript, we exclusively adopt this paradigm, since all the relevant uncertainties discussed here are epistemic in nature (relating to imperfect knowledge) and not aleatory (arising from some intrinsic source of 'randomness'). Thus, rather than considering 'the pdf of $S$' it is more correct to refer to 'my pdf of $S$' or perhaps 'our pdf for $S$' in the case that many researchers share a consensus view.

It should be noted that Bayesian probabilities, being personal in nature, are in general conditional on some personal 'background' set of beliefs of the researcher $\Omega_p$. Thus, $p(S)$ could be more precisely written as $p(S|\Omega_p)$. However, this background knowledge will usually be omitted for convenience, and conditioning will usually be explicitly included only when there is some specific information that may be considered particularly relevant (and which is not assumed to be widely known).

As we have seen, the question of (conditional) independence boils down to the question of whether $p(A|B \cap S)$ is equal to $p(A|S)$. Our discussion of the subjective nature of likelihood within the Bayesian probabilities should make it clear that there is not an objectively correct answer to this question, but rather it depends on the subjective view of the researcher in question. Posing the question presupposes that the researcher already has likelihoods $p(A|S)$ and $p(B|S)$ in mind, or else the observations $A$ and $B$ would not be considered useful evidence on $S$. Would knowing $B$ change their predictive distribution for $A$ (i.e. the likelihood $p(A|S)$)? If it would not, then $A$ and $B$ *are* conditionally independent given $S$, for this researcher. That is, if the researcher does not know how to use the additional information $B$ in order to better predict $A$, then $A$ and $B$ are conditionally independent to that researcher. Thus, ignorance implies independence. If, conversely, $B$ does provide helpful information in addition to $S$, then their improved prediction is the new likelihood function $p(A|B \cap S)$, which directly enables the joint likelihood $p(A \cap B|S)$ to also be created.

## 4 Model independence in the Bayesian framework

We now explore how this Bayesian framework can be applied to the question of model independence. We first consider the 'truth-centred' hypothesis which is perhaps most clearly presented by Tebaldi et al. (2005). In that work, the outputs of the models, $M_i$ (where $1 < i < n$ indexes the different models) are assumed to be samples from a multivariate Gaussian distribution centred on the truth $T$. The likelihood for each model $p(M_i|T)$ is therefore a Gaussian of the same width centred on the model outputs. The joint likelihood for multiple models is equal to the product of their individual likelihoods, which as we have seen above is equivalent to considering that the models are independent conditional on the truth. The joint likelihood will therefore be a Gaussian centred on the ensemble mean and its width will narrow in proportion to the square root of the number of models considered, which is the mathematical justification for the supposition that the ensemble mean will converge to the truth. As we have already mentioned, this behaviour is contradicted by analysis of the model outputs (Knutti et al., 2010). Thus, although such a definition of the concept of model independence could be presented in terms of the statistical definition of independence, it does not describe the behaviour of the models adequately because the models do in fact generally share common biases.

In light of this failure of the truth-centred approach, we now present two alternative interpretations of statistical independence that we believe could be more relevant and appropriate in application to the ensemble of climate models. We use CMIP3 here, rather than CMIP5, primarily in order that the ideas developed here can in the future be tested against a somewhat new sample, so as to defend against the risk of data mining.

Consider firstly the case where the outputs of a subset of the models which contributed to CMIP3 are labelled as $M_1, \ldots, M_n$, so as to conceal the underlying model names. If told that one of these models $M_1, \ldots, M_n$ was actually MIROC, say, then a researcher who was asked to identify which outputs came from this specific model and who did not have unusually detailed knowledge of this and other climate models would quite possibly assign uniform probabilities across these sets of outputs. Now consider how the situation would change if another set of outputs $M^*$ (not included in the original set) was provided, and identified as having been generated by the MRI model. If the same researcher was again asked to predict which of $M_1, \ldots, M_n$ was from MIROC, then their answer would either change, or it would not, depending on their beliefs concerning the relationship between these two climate models (which were contributed by neighbouring institutes in Japan and have some common origins). In the case that their answer did not change, this would imply that they considered the MRI and MIROC models to be independent, conditional on the unlabelled ensemble of model outputs. If, on the other hand, they thought MRI and MIROC were likely to be particularly similar among the ensemble of climate models (due either to the legacy of shared code or development methods) then it would be rational of them to assign higher probabilities to the sets of outputs that were closer to $M^*$ in some metric.

While the subjective nature of Bayesian probability precludes a definitive answer, we expect that for most researchers and most model pairs where there is no clear institutional or historical link, they will indeed believe the models to be independent in this manner (i.e., conditional on the unlabelled ensemble of outputs). Conversely, if the pair of models appear to differ in only some very limited manner, such as being different resolutions of the same underlying code (consider for example the T63 and T42 versions of CCMA which were submitted to CMIP3) then it might be sensible for a researcher to instead update

their prediction of the unknown model, increasing probabilities of outputs which were closer (according to some reasonable measure) to the named model, and with decreasing probabilities assigned to more distant outputs. The extent to which the probabilities are changed would be a direct indication of the strength of the dependence between the models, as judged by the researcher.

An alternative but similar approach can be formulated if, instead of using the discrete distribution of actual climate model outputs, we parameterise their distribution, for example as a multivariate Gaussian. If given the parameters of a Gaussian distribution based on the outputs of $M_1, \ldots, M_n$ (i.e. $\overline{M} = \sum_{i=1\ldots,n} M_i/n$ being the mean and $\sigma$ the standard deviation of the outputs), and asked to predict the outputs of MIROC (knowing it to be one of the constituent models), a researcher might reasonably decide that a reasonable answer would be to use this Gaussian directly as the predictive distribution. Additionally learning the outputs and true name of an additional model $M^*$ will leave their prediction unchanged if and only if the researcher thinks that this model is independent of MIROC, conditional on the ensemble distribution. If the researcher thinks that the model $M^*$ is related to MIROC, then they might plausibly modify their prediction, for example by shifting the original Gaussian towards $M^*$ in some way. A numerical example is provided in Section 4.1 below.

These approaches, we believe, encapsulate many of the same ideas as the model similarity analyses of Abramowitz and Gupta (2008); Knutti et al. (2013); Sanderson et al. (2015) and others. However, our approaches have the advantage that independence here can be defined in absolute terms (conditional on a clearly defined background knowledge) and is not merely a measure of relative difference. If a researcher does not know how to improve their prediction of a particular model, in light of being given a particular set of outputs from another named model, then this pair of models are in fact absolutely independent to them in statistical terms.

## 4.1 Example

To provide a concrete demonstration of the previous ideas, we analyse the models which contributed to the CMIP3 database. Several modelling centres contributed more than one model version and we expect, based on the existing literature such as Knutti et al. (2013), that these may be noticeably more similar to each other than two models from different randomly-selected centres would be. In total, we use the outputs of 25 climate model simulations, and analyse two-dimensional climato-logical fields of surface air temperature (TAS), precipitation (PREC) and sea level pressure (PSL) for their pre-industrial control simulations. We can identify 9 pairs of models where both were contributed by the same institute and use these as examples of models that we expect to show dependency, but note that this approach does not make use of any detailed knowledge of model development or shared code and other researchers might make different choices if asked to predict dependence among the ensemble.

We use as a simple distance metric the area-weighted root mean square difference between the climatological data fields (of commensurate variables) after regridding to a common 5 degree Cartesian grid. For example, randomly selected models from the ensemble have an area-weighted RMS difference of around 3°C. Given the model fields — or even just their pairwise RMS differences — it would surely be difficult for most researchers to identify with any confidence which field came from a specific model such as CSIRO3.0, and if asked to provide a probabilistic prediction, they might reasonably assign uniform

probabilities across the set. However, if the researcher is then given the outputs of a new model $M^*$ and told that it was in fact CSIRO3.5, it would now be reasonable to expect that CSIRO3.0 was more likely to be one of its near neighbours rather than relatively distant from it, under the assumption that the changes arising from the development between these model versions were relatively modest. A simple way to account for this expected similarity, in terms of formulating a probabilistic prediction

for the outputs of CSIRO3.0, would be to assign probabilities to the unnamed sets of outputs, in some way such that the probability decreases with distance from CSIRO3.5. By way of demonstration, we order the unnamed models by increasing distance from CSIRO3.5 and assign them probabilities that decrease proportional to the sequence $1/1, 1/2, 1/3, \ldots, 1/n$. The choice of this particular sequence was of course highly subjective and many different distributions could have been used instead.

A researcher who applied this probabilistic strategy to each of the 9 pairs of models identified as coming from the same centre, would assign a typical (geometric mean) probability of around 0.09 to the correct field of outputs, when averaged over all model pairs and over the three types of fields TAS, PREC and PSL. The naive uniform distribution would in contrast only assign a probability of $1/24 \simeq 0.04$ to the correct field. Thus, taking account of the shared origins can typically increase the probability of a correct prediction by a factor of more than two, and we may conclude that models from the same institute

are not independent, conditional on knowing the pairwise distances between their outputs. This is of course little more than a mathematical interpretation of the similarities noted by Masson and Knutti (2011) and others. Thus, the result is not surprising, but we believe it is worthwhile to demonstrate how those earlier empirical investigations can be explained and expressed directly in terms of statistical independence. The results for each pair of related climate models, and for each of the three climate fields considered here, are presented graphically in Figure 1.

Similar results can be obtained when the analysis is performed in parametric terms, when rather than using the sets of model outputs, only a statistical summary of the ensemble of outputs is provided in the form of multivariate Gaussian approximation to their distribution $N(\overline{M}, \sigma)$ where $\overline{M} = \sum_i M_i/n$ is the ensemble mean and $\sigma$ is the standard deviation of the distribution. In this case, we consider a researcher who is asked to predict the location of an additional model $M_{n+1}$. A natural prediction is simply the distribution $N(\overline{M}, \sigma)$. The question of dependence then rests on whether, when told the location of a plausibly

related model $M_j$ already contained in the ensemble, the researcher changes their prediction. One interesting detail to note is that for most model pairs $(M_j, M_{n+1})$ provided by a single modelling centre, the outputs of $M_j$ actually provide a marginally worse prediction of $M_{n+1}$ (in the sense of being further away) than the ensemble mean $\overline{M}$ does. However, this very small increase in distance suggests that an interpolation almost half-way from the ensemble mean to $M_j$ might provide a better prediction still, and we find that this is indeed the case. Using $M'_{n+1} = 0.6\overline{M} + 0.4M_j$ as a predictor for $M_{n+1}$ generates a

measurably lower prediction error, typically by 10% or so across the three data fields used, than the original ensemble mean $\overline{M}$ did. Therefore, the original prediction of $N(\overline{M}, \sigma)$ can be replaced by $N(M'_{n+1}, 0.9 \times \sigma)$, to give a better prediction of the unknown model $M_{n+1}$. This result demonstrates empirically and numerically that two models contributed by a single research centre are not conditionally independent given $\overline{M}$ and $\sigma$. These results are also presented graphically in Figure 1.

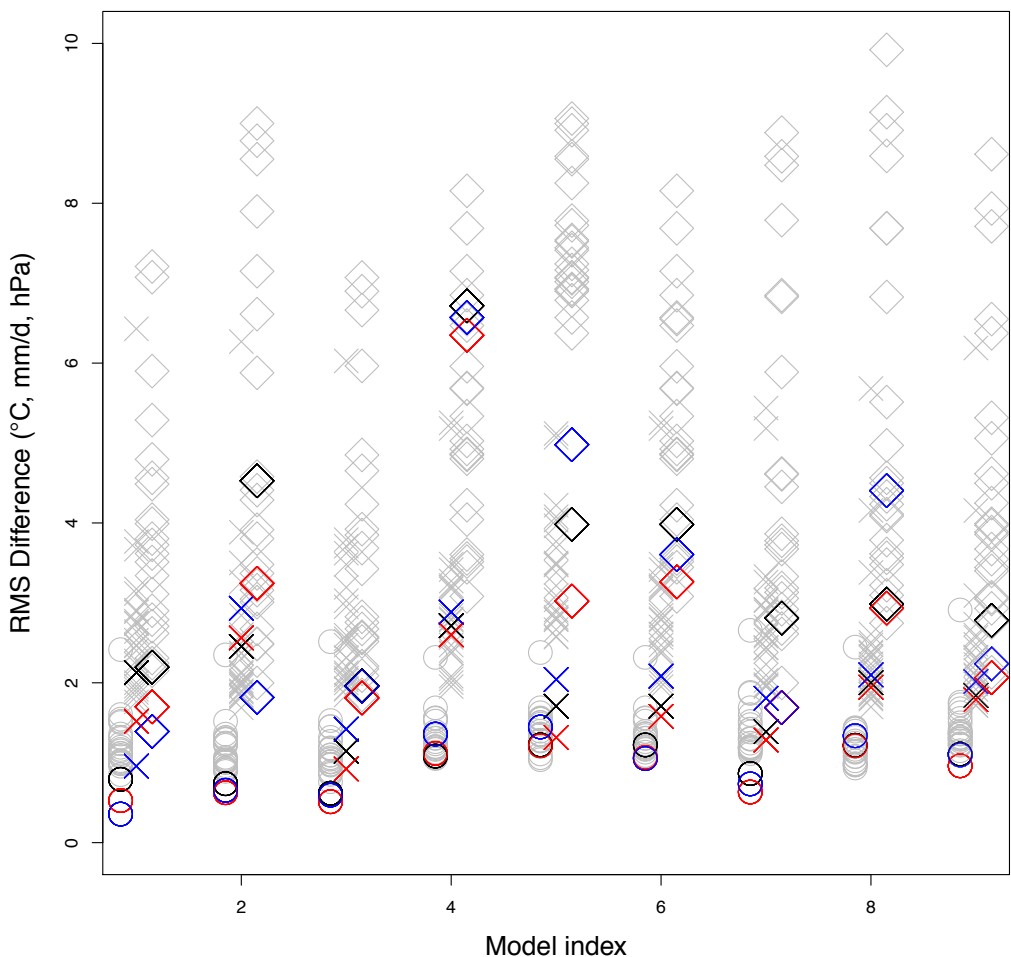

**Figure 1.** Analysis of CMIP3 models. The x-axis indexes the 9 model pairs being considered. Crosses represent TAS, circles PREC and diamonds PSL. Grey symbols indicate RMS distances of all models from the target model of the pair. Black symbols indicate distance of target model from mean of residual ensemble, blue symbols indicate distance of target model from plausibly related model, and red indicates distance of target model from interpolated prediction described in the text. The blue symbols being closer to zero than almost all grey symbols shows that related models are typically closer together than randomly selected models, and comparing red and black symbols shows that the interpolation improves as a predictor over the ensemble mean in almost all individual cases, and overall.

## 4.2 Accounting for model dependence via weighting

A natural question to ask is whether some weighting scheme could be developed to account for model dependence of this type. If we anticipate that a pair of models will be particularly similar, then including both in the ensemble without downweighting either of them will tend to shift the ensemble mean towards this pair of models. The correct weight to prevent this can easily be calculated according to the interpolation formula in the following manner. If we anticipate that a particular model $M_j$ will help to predict a new model $M_{n+1}$ via an interpolated prediction $M'_{n+1} = (1-\alpha)\overline{M} + \alpha M_j$ for some coefficient $0 < \alpha \leq 1$, then adding $M_{n+1}$ to the ensemble without any adjustment to weights (i.e. with all model weights equal by default) will result in an *a priori* expectation that the ensemble mean will be shifted towards $M_j$, with the effect being stronger the closer $\alpha$ is to one. One simple approach to counteract this effect would be to discard the candidate new model, effectively giving it a weight of zero. However, the resulting ensemble would be sensitive to the order in which models are added, and the symmetry of the dependence relationship suggests that it would be more reasonable to apply an equal weight to each model in the dependent pair. If an equal weight of $1/(1+\alpha)$ is applied to both models (relative to unit weighting on the other models), then the prior expectation will be that the ensemble mean is unchanged by the inclusion of the additional model. Perhaps the simplest way to show this is to start from the identity that for the original ensemble

$$\sum_i (M_i - \overline{M}) = 0 \tag{5}$$

due to the definition of the mean. If the additional model has an expected output of $M'_{n+1} = (1-\alpha)\overline{M} + \alpha M_j$ and we apply the same weight $w$ to both models $M_j$ and $M_{n+1}$ then our prior expectation for the equivalent sum over the weighted, augmented ensemble is given by

$$\sum_{i \neq j} (M_i - \overline{M}) + w(M_j - \overline{M}) + w((1-\alpha)\overline{M} + \alpha M_j - M), \tag{6}$$

which simplifies to

$$\sum_{i \neq j} (M_i - \overline{M}) + w(1+\alpha)(M_j - \overline{M}) - w(1+\alpha). \tag{7}$$

This sum equals zero when $w = 1/(1+\alpha)$.

For example, if a second identical replicate of an existing model were to be contributed to the ensemble (in which case $\alpha = 1$) then both models will receive a weight of 0.5, precisely cancelling out the duplication. In the numerical example presented above, we have chosen $\alpha = 0.4$ and thus the appropriate weight would be $1/1.4 \simeq 0.7$. Weighting the models will not be expected *a priori* to affect ensemble spread, as we have no expectation that the dependent models are systematically closer to, or further away from, the ensemble mean when compared to the rest of the ensemble. The effect of weighting on ensemble mean performance is also expected to be very small as the change in effective ensemble size (which can be defined as $1/\sum_i w_i^2$ where $\sum w_i = 1$ are the relative weights) will be modest. If we have an initial ensemble of say 15 independent models and then 8 of these models are effectively assigned relatively higher weights of 1.4 by addition of near-replicates to the ensemble, then

the effective ensemble size will only decrease from the original 15, to a new value of 14.6. This is a negligible difference that cannot be expected to affect ensemble performance in any measurable way. Figure 3c and 3d of Knutti et al. (2010) shows that the typical performance of a randomly-selected sub-ensemble of say 20 models is only very marginally worse than the full set of 23 used in that paper. However, if a future CMIP ensemble was dominated by a large number of near-replicates of a small subset of models, then this issue would undoubtedly become more important.

## 5 Discussion

We have presented a coherent statistical framework for understanding model independence, and demonstrated how this framework can be applied in practice. Climate models cannot sensibly be considered independent estimates of reality, but fortunately this strong assumption is not required in order to make use of them. A more plausible, though still optimistic, assumption, might be to interpret the ensemble as merely constituting independent samples of a distribution which represents our collective understanding of the climate system. This assumption is challenged by the near-replication of some climate models within the ensemble, and therefore sub-sampling or re-weighting the ensemble might be able to improve its usefulness. We have shown how the statistical definition of (conditional) independence can apply and how it helps in defining independence in a quantifiable and testable manner.

The definition we have presented is certainly not the only possible one and we expect that others may be able to suggest improvements within this framework. For instance, experts with knowledge of the model structures might be able to predict more detailed similarities between the outputs of model pairs. Moreover, there is no requirement that, in applying our principles, a researcher would use the most naive ignorant prediction, of uniform probabilities across the ensemble of outputs, or the Gaussian summary of the distribution, respectively, as their predictions of the target model. However, our result here is sufficient to illustrate how the concept of statistical independence can be directly applied in a quantitative mathematical sense to the question of model independence, while encapsulating much of what is discussed in the literature.

An important point to note is that this interpretation of independence is entirely unrelated to model and indeed ensemble performance (eg Reichler and Kim, 2008; Annan and Hargreaves, 2010). Here we consider these questions to be separate topics, which require study in their own right. Reality (e.g. observations of the real climate system) does not enter into any of the calculations or definitions above. Thus, the two concepts of performance and independence as used here are entirely unrelated. It remains a challenge to develop some useful interpretation of (conditional) independence which *does* use real data and which is informative regarding both model performance and pairwise similarity. However, the definition as presented here does have obvious applications in terms of interpreting and using the model ensemble. It suggests that we may be able to usefully reduce the full CMIP ensemble to a set which are independent (conditional on the ensemble statistics, as above). This will provide a smaller set of models for analysis and use in downstream applications including downscaling to higher resolution regional simulations of climate change. This is likely to be increasingly important and necessary given the heterogenous nature of simulations which will likely be submitted to future CMIP databases. An additional point which should not be overlooked is that the numerical example presented here was undertaken purely in terms of the modern climatologies of the models, and

does not consider future climate changes. However, the underlying principles of independence do apply more broadly to any consideration of model outputs, and the conclusions reached may be different depending on the data sets used.

While the question of model similarity and ensemble member selection has already been considered by others (eg Sanderson et al., 2015), the work here provides a more clear-cut definition of what it means to be independent, which is directly testable. If researchers can demonstrate dependence (in terms of an improved prediction of model outputs as illustrated here) then independence is violated, and if not, it may be reasonably assumed. Another important difference between the approach presented here, and that of many other authors, is that independence is determined *a priori* in terms of the anticipated outputs of the models, rather than *a posteriori* in light of the model outputs. Pairwise similarity between model outputs may arise through convergence of different approaches to understanding the climate system, and not merely through copying of ideas, and this would not indicate any dependence as defined here. In fact, one pair of models which exhibit unusually similar temperature fields in our analysis consists of the model from CNRM and one of the GFDL models, which do not share any particularly obvious relationship. We do not believe that coincidentally similar behaviour should be penalised by downweighting of these models, as it may represent a true 'emergent constraint' on system behaviour. An obvious future test of our ideas would be to apply this analysis to the CMIP5 and CMIP6 ensembles of climate models, to check whether the interpolation and dependence ideas presented here apply generally to ensembles of climate models rather than being an example of over-enthusiastic data mining.

# Part II

# Independence of constraints on climate system behaviour

In Part I, we discussed how the concept of independence applies to the sets of models which form the CMIP ensembles of opportunity. In Part II, we discuss estimation of climate sensitivity, although the principles presented here apply more generally to observational constraints on climate system behaviour. While initially it may seem that this topic has little in common with that of Part I, we will show how the concept of probabilistic independence also relates directly to this question. Thus, the probabilistic background of Section 3 is directly relevant and applicable here.

## 6 The literature concerning observational constraints on the climate sensitivity

The magnitude of the equilibrium climate sensitivity $S$ (the globally-averaged equilibrium temperature response to a doubling of atmospheric $CO_2$) has long been one of the fundamental questions of climate change research (Charney et al., 1979). A wide range of approaches have been presented which attempt to estimate this number. Most commonly, a Bayesian approach is used in which some prior estimate is updated by means of an observationally-based likelihood function to form a posterior estimate. The observations frequently relate to the warming observed during the instrumental period (which we refer to for convenience as the 20th century, although the relevant observational data available does extend into the 19th and 21st centuries) (Tol and

De Vos, 1998; Forest et al., 2006; Skeie et al., 2014), but analyses have also been presented which use longer-term climate changes seen during the paleoclimate record (Annan et al., 2005; Köhler et al., 2010), or short-term variations seen at seasonal to interannual time scales (Wigley et al., 2005; Knutti et al., 2006). In each case however, the observations are not a direct measure of the sensitivity $S$ per se but must be related to it through the use of a climate model or models, which may be simple or complex. Collins et al. (2013, Box 12.2) and Annan (2015) survey and discuss some recent analyses which use a variety of observational data sets and modelling approaches, and Rohling et al. (2012) covers the paleoclimate field in some detail.

The question naturally arises as to whether these different constraints could, and should, be synthesised. In most of the Bayesian analyses, the prior is typically chosen to be vague, though there is some debate concerning this choice (Annan and Hargreaves, 2011a; Lewis, 2014). Irrespective of the choice of prior, the posterior after updating with observations is typically substantially narrower. One might reasonably wonder what the results would look like if this resulting posterior was then used as the prior in a new analysis in which it was updated by a *different* data set. This question was first explicitly raised by Annan and Hargreaves (2006), who made an assumption of independence between the constraints and thus implemented a straightforward process of sequential updating using Equation 4 which resulted in a substantially tighter constraint than had previously been obtained. Hegerl et al. (2006) similarly updated a posterior arising from an estimate based on the 20th century warming, with a separate data set relating to climate changes over earlier centuries. However, the validity of these analyses is not immediately obvious, as the independence of different constraints has not been clearly explained or demonstrated. Nevertheless, we always expect to learn from new observations (Lindley, 1956), so it is reasonable to expect that an analysis which accounts for multiple lines of evidence will generate a more precise and reliable result than analyses that do not. It is therefore surprising that there has been very little discussion of this topic in the climate science literature, and very few recent attempts to combine diverse data sources, although this topic is now receiving some fresh attention (Stevens et al., 2016).

## 7   Independence of constraints in the Bayesian context

It should be clear from the discussion in Section 3 that the concept of independence in relation to multiple constraints on the equilibrium climate sensitivity $S$ is more precisely expressed as conditional independence of these constraints given $S$. The issue is whether it is valid to replace the term $p(A \cap B|S)$ in Equation 3 with $p(A|S)p(B|S)$ to form Equation 4, or equivalently whether $p(B|S \cap A) = p(B|S)$. This is essentially the same concept as the 'truth-centred' approach to model independence discussed briefly in Section 4, although the skewed and asymmetric forms of general likelihood functions means that it is not necessarily appropriate to think of them as being centred on the true value of $S$.

In Section 3.1, we argued that ignorance of any dependency implies independence. Given a likelihood $p(A|S)$ we ask ourselves, how can we change this by additionally including $B$ to form $p(A|S \cap B)$? If the answer is that $B$ provides no additional information regarding $A$ (conditional on knowing $S$), then $A$ and $B$ are conditionally independent given $S$. This answer may seem a little unsatisfactory, as it relies on a dogmatically subjectivist and personal interpretation of probability. While we emphasise that Bayesian probability is at its heart a fundamentally subjective concept, it is quite usual to use numerical or mathematical models as a tool to represent and understand our uncertainties.

While the subjective nature of Bayesian priors (i.e. $p(S) = p(S|\Omega_p)$ where $\Omega_p$ is the researcher's personal background knowledge) has been regularly discussed in the literature, it is less widely appreciated that the likelihood $p(A|S) = p(A|\Omega_p \cap S)$ is also a fundamentally subjective concept within the Bayesian paradigm. Even if $S$ is a well-defined property of the real world (which is not always immediately clear when $S$ is defined in sufficiently abstract terms), there is then no alternative world in which $S$ takes a different value, with which we could check to see which events take place in this case. Therefore, while the likelihood should give a reasonable prediction of the evidence $A$ when the correct value of $S$ is used, there is no objective constraint or check on what the likelihood should predict for some alternative incorrect choice of $S$. The only practical way in which a likelihood can be constructed is via some model which allows $S$ to vary, either as an explicit parameter in a simple model, or perhaps as an emergent property of a more complex model which includes multiple sources of uncertainty. There can be no 'correct' way to vary $S$, again because there is no world in which $S$ takes a different value against which to validate our choices. Within the Bayesian paradigm, therefore, the likelihood can only reflect the researcher's subjective beliefs and modelling choices rather than any physical truth. Different models will in principle lead to different likelihoods, though in practice there may be a reasonable level of agreement between researchers.

## 7.1   Example

Here we explore these ideas in a little more detail, to illustrate how it is possible to provide a credible basis for what are fundamentally subjective judgements. Typically, a likelihood $p(A|S)$ is generated not as a purely subjective matter of belief, but instead justified via a model or ensemble of models. For example, if the equilibrium sensitivity is varied across an ensemble of energy balance models (along with other input parameters: $S$ here may be used as a shorthand for a vector of relevant uncertainties) then we will find that in simulations of the 20th century, the warming observed will vary across the ensemble. This can then be used as the basis for the likelihood function (eg Tol and De Vos, 1998; Forest et al., 2006; Skeie et al., 2014). Similarly, for another observable $B$ such as the cooling during the Last Glacial Maximum (LGM), which may require another set of simulations using the same ensemble. We now outline how it is possible to test whether $B$ is conditionally independent of $A$ given $S$, in the context of this model.

A simple example is used to illustrate the point. We use a zero dimensional energy balance model to simulate the climate changes of both the 20th century and the LGM. For simplicity, we only consider a subset of the relevant uncertain parameters: the equilibrium sensitivity $S$, the planetary effective heat capacity $C$, the uncertainties in radiative forcing due to aerosol forcing over the 20th century $F$, and atmospheric dust and the large ice sheets which existed during the Last Glacial Maximum, $D$ and $I$ respectively.

For the warming of the 20th century, we assume the total forcing $G = G(t)$ follows a linear forcing ramp from 0 in 1900 to $2 - F$ in 2000 (using a value of $2Wm^{-2}$ to approximately represent the sum of all other forcings other than aerosols, which

are dominated by greenhouse gases). We simulate the climate change with the zero dimensional energy balance model which satisfies the equation

$$C \cdot dT/dt = G(t) - T(t) \times 3.7/S, \tag{8}$$

where $T(t)$ is the temperature anomaly (relative to 1900) at time $t$. The radiative forcing due to a doubling of $CO_2$ is taken to be $3.7 Wm^{-2}$. Our first observable $A$, the change in global mean surface air temperature over the 20th century as estimated by the linear trend over this interval. During the LGM, the climate can be assumed to be at a quasi-equilibrium and thus the planetary heat capacity (which moderates transient changes) can be ignored. The equilibrium temperature anomaly $B$ during this period is calculated as

$$B = (3 + D + I) \times S/3.7 \tag{9}$$

where the total forcing $3 + D + I$ is the sum of greenhouse gases ($3 Wm^{-2}$), the uncertain dust forcing ($D$) and the uncertain effective forcing of the ice sheet ($I$) respectively. The ice sheet forcing uncertainty term used here implicitly accounts for the nonlinearity of how this combines with the other forcings. For simplicity, we do not consider observational uncertainties for either the LGM or 20th century temperature changes, though accounting for these would be straightforward. We use the following priors which are all taken to be either uniform distributions $U[,]$ or Gaussian $N(,)$:

$S \sim U[0.5, 6]$

$C \sim U[10, 30]$

$F \sim N[1, 0.5]$

$D \sim N[1, 0.5]$

$I \sim N[3, 1]$

A plot of the simulated 20th century warming $A$ versus sensitivity $S$ is shown in Figure 2a, together with a linear regression fit to these data. This relationship shown demonstrates the basis for a likelihood function $p(A|S)$: for any specified sensitivity we can predict the resulting temperature using the regression line (albeit with uncertainty), and therefore we can calculate how the probability of any specific warming $A$ varies with $S$. In this example, the linear regression provides a good fit to the data, though the uncertainty clearly grows towards larger sensitivity values. Similarly, the LGM cooling $B$ is also linked to $S$ (Figure 2b) and this relationship can be used as the basis for a likelihood function $p(B|S)$.

By construction, we already know that the two constraints are independent given $S$, since the other uncertain parameters that relate to each observations are disjoint. However, if we did not know this analytically *a priori* but were instead merely

able to use the model as a black box, we could check for the independence of two sets of constraining evidence $A$ and $B$ in the following manner. Firstly we would form the likelihood $p(A|S)$ as above, and use this together with the known value of $S$ for each ensemble member to generate a mean prediction (which we denote $A'$) of the observation for each. On comparing to the actual observed value $A$ for each ensemble member, there will typically be a residual $(A - A')$ between the predicted and observed values, the magnitude of which indicate the limited information which $S$ provides concerning $A$. We can now explore whether an additional observable $B$ is informative regarding these residuals, i.e., whether it exhibits any systematic relationship with them. If it does not, then we may reasonably conclude that $B$ provides no additional information on, and is conditionally independent of, $A$ given $S$. Conversely, if $B$ is informative regarding the residuals, then this is proof that it is not independent of $A$.

In the context of our example, we first create an ensemble with an arbitrary but fixed value of $S = 3.5°$C, say, and simulate both the 20th century warming and the LGM state for each member of this ensemble. The likelihood function arising from Figure 2a gives us a predicted warming of $A' = 0.85°$C (with uncertainty of $0.4°$C) for these ensemble simulations. We now check the prediction errors to see if they exhibit any relationship with $B$. Figure 2c indicates that they do not, with the regression coefficients being insignificantly different from zero. The conclusion is that the additional knowledge of $B$, once the sensitivity $S$ is known to be $3.5°$C, does not provide any additional help in predicting $A$. $A$ and $B$ are therefore independent, conditional on $S = 3.5$C. This experiment can be repeated for as many different values of $S$ as is desired, and the same negative result will be found. This is of course not surprising, as the model has been constructed in this way.

We now make a small change to the model, and substitute $D$ with $F$ in Equation 9 to obtain $B = (3 + F + I) \times S/3.7$. This modified model now makes the assumption that the magnitude of effective dust forcing at the LGM is the same as that of the aerosol forcing during the 20th century. This is of course again a very simplistic approach but it is not completely unreasonable to assume a link of some sort, as both forcings relate to the effects of condensation nuclei on clouds. Importantly, the univariate likelihood functions $p(A|S)$ and $p(B|S)$ are unchanged by this substitution, as $D$ and $F$ are identically distributed. Therefore, we can generate the same prediction for $A$, conditional on a known $S = 3.5°$C. However, with this change to the model, the prediction errors are now strongly correlated with $B$, as is shown in Figure 2d. Therefore, a new distribution function $p(A|S \cap B)$ can be created which makes a more precise prediction of $A$ given knowledge of both $S$ and $B$. Thus it can be diagnosed from the model outputs alone, without direct knowledge of the model's internal structure, that $A$ and $B$ are not independent conditional on $S$. This result is of course easily interpreted in terms of the known model structure: for a given sensitivity, a smaller than expected cooling at the LGM suggests a low dust/aerosol forcing, which then implies that the 20th century warming will be greater than would be expected from knowledge of sensitivity alone.

The linear regressions are not necessarily the best way to represent a relationship that may in practice be more complex. However, such an approach may be expected to capture any first-order effect. The central point of these numerical experiments is to demonstrate that this dependence can in principle be diagnosed from model outputs directly, without the need for detailed knowledge or understanding of causal relationships embedded in the model structure. Furthermore, a conditional likelihood $p(A|S \cap B)$ can subsequently be generated from the ensemble outputs. This then enables us to generate the joint likelihood $p(A \cap B|S) = p(A|S \cap B)p(B|S)$ as required for a Bayesian inversion.

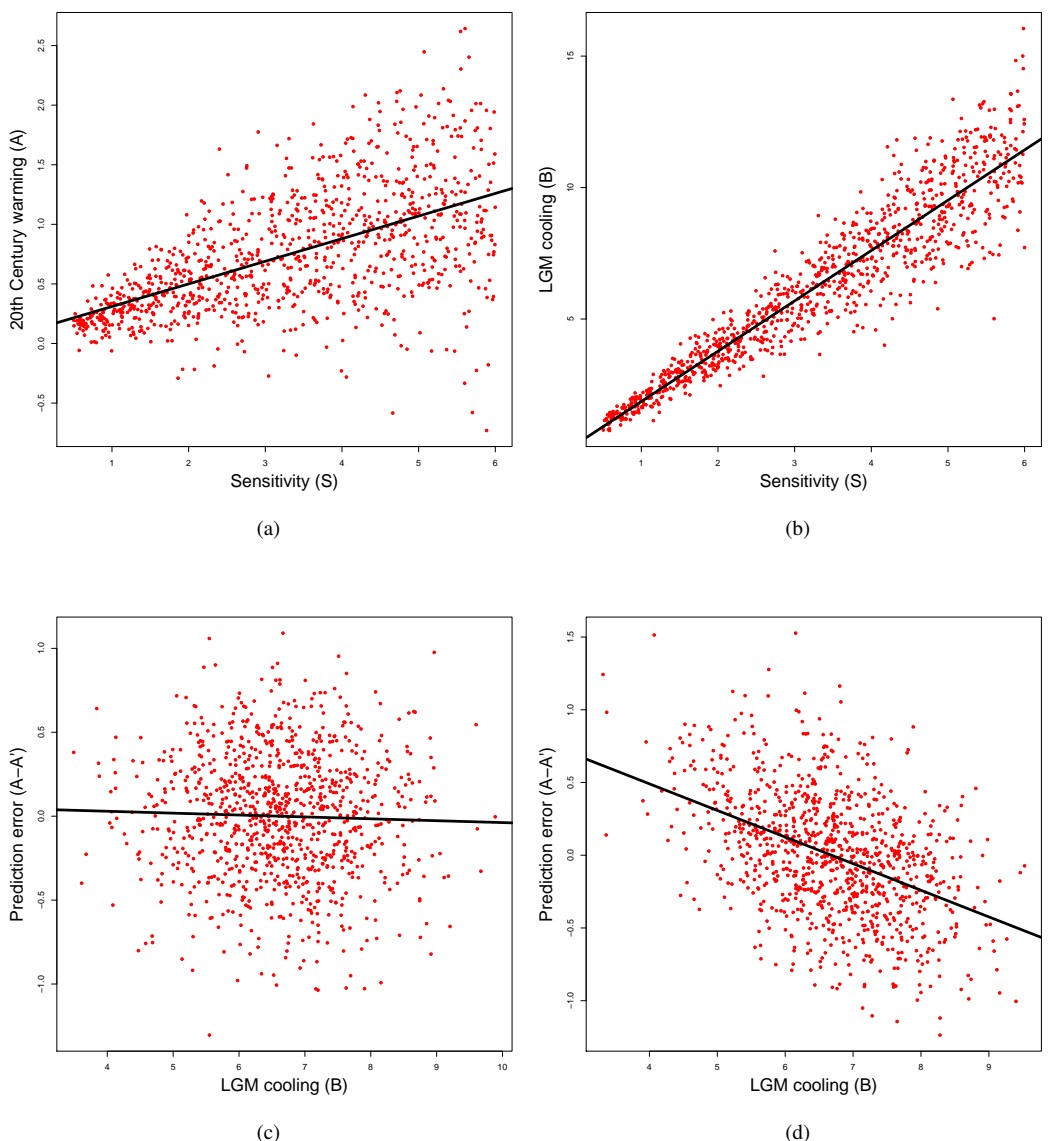

**Figure 2.** Outputs of ensemble simulations (red dots) and linear regression fits (black lines): (a) 20th Century warming ($A$) versus equilibrium sensitivity (b) LGM cooling ($B$) versus equilibrium sensitivity (c) 20th century prediction residuals ($A - A'$) versus LGM cooling ($B$), independent case (d) 20th century prediction residuals ($A - A'$) versus LGM cooling ($B$), dependent case.

Such analyses may be impractical for the outputs of small ensembles such as those arising from the CMIP multi-model experiments which explore structural uncertainties. However, they may well be plausible for larger ensembles where parameters are varied within a single model structure. The key requirement is that the simulations relating to different observables are performed with the same model, in order that any dependence between constraints can be explored. The results obtained will of course depend on the model used, but this is as expected: the likelihood is not a property of reality, but rather, a consequence of the modelling assumptions, as was discussed in Section 7.

## 8  Summary of Part 2

The question of how to combine multiple constraints on climate sensitivity has been occasionally raised, but more commonly ignored, in analyses of this parameter. It is well-known that combining constraints should lead to more confident conclusions, but the difficulty of accounting for possible dependency appears to have widely discouraged researchers from attempting this (Collins et al., 2013, Box 12.2). This situation may start to change (e.g. Stevens et al., 2016), and we hope that the analysis presented here will encourage others to consider the question of dependence more directly. In particular, we have argued that independence is fundamentally a subjective matter, but we have also shown how it may in principle be diagnosed from an ensemble of models which purports to represent our subjective uncertainties. A more widespread use of model ensembles which simulate multiple observationally-constrained periods (such as both modern and paleoclimate periods) may enable more progress to be made.

## 9  Conclusions

We have discussed and presented a coherent statistical framework for understanding independence, and explained how this applies in two distinct applications. Climate models cannot sensibly be considered independent estimates of reality, but fortunately this strong assumption is not required in order to make use of them. A more plausible, though still optimistic, assumption, might be to interpret the ensemble as merely constituting independent samples of a distribution which represents our collective understanding of the climate system. This assumption is challenged by the near-replication of some climate models within the ensemble, and therefore re-weighting or sub-sampling the ensemble could improve its usefulness. We have shown how the statistical definition of (conditional) independence can apply and how it helps in defining independence in a quantifiable manner. The definition we have presented is certainly not the only possible one and we expect that others may be able to suggest improvements within this framework.

When considering the use of observational evidence in constraining climate system behaviour (including the specific example of the equilibrium climate sensitivity), observational uncertainties themselves can generally be regarded as independent. However, the independence of the resulting likelihood functions is not so immediately clear, as it typically also rests on a number of modelling assumptions and uncertainties. Here we have shown how the question of independence can be readily

interpreted and understood in terms of the conditional prediction of observations. These ideas may be useful in the design and analysis of ensemble experiments underpinning the analysis of observational constraints.

While our examples do not provide complete solutions to the questions raised, we have shown how the statistical framework can be usefully applied. Further, we see little prospect for progress to be made unless it is underpinned by a rigorous mathematical framework. Therefore, we hope that other researchers will be able to make use of these ideas in their future work.

*Author contributions.* Both authors contributed to the research and writing.

*Acknowledgements.* We acknowledge the modelling groups, the Program for Climate Model Diagnosis and Intercomparison (PCMDI) and the WCRP's Working Group on Coupled Modelling (WGCM) for their roles in making available the WCRP CMIP3 multi-model dataset. Support of this dataset is provided by the Office of Science, U.S. Department of Energy. We are also particularly grateful for the many helpful suggestions made both by the reviewers, and by many participants at a recent meeting at NCAR concerning this topic.

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
