# Peer review of "On the meaning of independence in climate science"

_Earth System Dynamics, 2016_

## Short Comment (SC1) · 5 Sep 2016

N. Herger

nadja.herger@student.unsw.edu.au

This is an interesting manuscript which highlights the lack of an agreed-on definition of "independence" in climate science. As stated by the authors, independence has a statistical definition which is encouraged to be used more often. Two examples are given where this statistical framework is applied. I have a few comments and suggestions.

Section 2.1: I was expecting to read something about the Replicate Earth paradigm that C. Bishop and G. Abramowitz were working on (Climate Dynamics, 2013 and Journal of Climate, 2015). I think their idea of ensemble interpretation would only add to the summary of literature on model independence. It assigns weights to models and transforms the whole ensemble to be more "Replicate Earth"-like, and thus independent.

P8, L27-28: The authors use climatological fields of CMIP3 model runs. How many

years were used for that? Depending on the length it might be that the RMSE values (as shown in Figure 1) are influenced by internal variability.

P10, L12-14: The author's interpretation of independence is entirely unrelated to model performance. Does that mean that it is not relevant to know where the models lie with respect to the observation(s) in some kind of projected model space? Most publications I have seen on this topic are only able to tell if a specific model is independent if their relative "position" to other models and the observation are known.

P11, L23: I'd suggest to introduce the abbreviation LGM here as this abbreviation is then later used on P12, L3.

Figure 2: There seems to be a mistake in the caption. The authors wrote that 2c) is the "dependent" whereas 2d) is the "independent" case. Shouldn't it be the other way around? The main text also suggests that those two words should be swapped.

Section 3.3: Couldn't the correlation between two climate variables come from shared model biases ("model dependence") rather than physical laws? I guess this is the danger of diagnosing dependence from model outputs directly. The expectation to find an error correlation of 0.5 for perfectly independent models might help gauge the degree to which two climate variables are physically correlated.

---

## Author Comment (AC1) · 8 Sep 2016

J. Annan and J. Hargreaves

jdannan@blueskiesresearch.org.uk

Thank you for the useful comments. Our point-by-point replies are as follows.

2.1 Failing to refer to the work of Bishop and Abramowitz was an oversight, as we didn't realise how substantially their approach differed from the cited Abramowitz and Gupta. However, this does not materially affect our argument.

p8 We used 30 year means, which certainly includes some internal variability but does allow for adequate discrimination of the models' equilibrium climates.

p10 We agree that in any overall assessment of model performance, the relationship of model output to observations of the real climate system is crucial. Our goal here was to emphasise that in the definition of independence presented here, observations

are not used and thus it is purely an indication of inter-model relationships. It remains unclear to us whether the concept of independence should or can usefully be defined so as to also account for model performance in relationship to observations.

p11 agreed

Fig 2 Thank you for spotting this error. The captions are indeed swapped.

3.3 Note that the correlation here is measured between model inputs and climatological outputs, as parameters vary, and not over time series as in Bishop and Abramowitz. Therefore model biases (relative to observations) do not play a role. Whether the relationship between parameters and outputs embodied in the model is correct or not is an essentially unfalsifiable proposition, as outlined earlier in the text. The model is considered a representation of the researcher's beliefs.

―――――――――――――――

---

## Referee Comment (RC1) · B. Sanderson (Referee) · 17 Sep 2016

The submitted paper discusses a framework for the consideration of climate model independence. In their discussion, the authors note that recent papers discussing climate model interdependencies have not identified a clear definition of what is meant by the term, and furthermore there is no clear relationship between the vague concept of known structural relationships between climate models and the formal statistical definition of independent, or conditionally independent data as it pertains to our confidence in a given projection of the future.

The paper raises some important questions, but its focus is currently mixed between two concepts which I am not yet convinced should be considered in the same paper. Section 3.2 presents a framework for considering the co-dependency of climate models

in the CMIP archive, while Section 3.3 provides a framework for assessing whether given model outputs represent independent constraints on a future projection given an ensemble whose members have an unknown structure. Although I understand that the authors are trying to make the point that the concept of conditional independence of data is universal from a Bayesian perspective – I don't yet feel like the manuscript marries these two concepts. Specifically, it is not yet clear what the aspiring Bayesian should do with information on model codependency once it has been confirmed.

The existing literature on this topic has to date been focused on analyses which relate inter-model distances to expected model relationships. As such, models exhibiting similar bias patterns have thus been considered potentially codependent and could be potentially down-weighted to account for this replication. The authors of the present study argue that such an approach could potentially erroneously downweight models which converge (through independent logic) to a similar 'truthful' solution.

In their consideration of model independence, they begin by considering why the 'truth-centered' worldview is inappropriate for an ensemble such as CMIP by noting that if the output of one model at a given location is known (as well as the true model), then this should change our expectation of another unknown model because we expect models to have biases of the same sign. Even if this is true, this is not a proof of the non truth+error nature of the ensemble, rather it is an assertion that we believe that the ensemble has a common bias in its present day state.

The authors then present their worldview, which is that codependent behavior should be expected in some cases from the archive (where there are clear examples of common code, or where models arise from the same institutions), and that the data from the models can be used to test that suspected dependency. In an example using the CMIP3 archive of historical simulations, the authors show that Euclidean distance between models from the same institution is less than the distance between two random models in the archive, and this can be seen as evidence of an a priori belief that the results of these models are therefore not independent. Furthermore, this knowledge
can be used to produce a better prediction of the state of an unseen model from an institution which already has a model in the archive (compared a prediction based on the existing ensemble mean).

What the authors should make more clear is what should be done with this information once it has been acquired. Once we suspect that two models are somewhat related, and the data has confirmed this, what is the appropriate course of action to use this information? The authors could follow up with a case study (perhaps broadly along the lines of Tebaldi, 2005) of how a simple projection informed by CMIP would change if the assumption of independent, truth centered models was dropped and replaced with assumptions informed by their analysis.

Another more general concern is this philosophy raises as many questions as it potentially solves. Firstly, there is obviously a requirement to have a prior assessment of which models should be independent, in order to use the ensemble data to test that assessment. Models from the same institution represent one source of potential codependency, but the cross-pollination of model components grows ever more complex with each generation of CMIP archive.

The authors use CMIP3, implying that they consider successive generations of the archive to be functionally equivalent for conceptual arguments such as the one they present. However, in some ways, CMIP3 represents a simpler world - CMIP5 (and likely more-so CMIP6) exhibits a more complicated mélange as components and entire models are ported from one institution to another. Tabulating and formulating this information into expectations of cross-model interdependency is itself a gargantuan task so it must be considered whether it is a practical recommendation to consider the potential independence of every model pair in a rapidly expanding and incestuous model family.

The authors rightly acknowledge that an a posteriori downweighting of models with similar biases could potentially downweight our most likely models if this was the only

action we took. However, I personally see the type of weighting done for interdependency and emergent constraints as somewhat tangential. The analyses presented thus far in the literature have found that the clearest evidence of common structure from model output comes from the consideration of distances in a high-dimensional space. As more unrelated variables are added to a metric, if two models continue to exhibit similar biases, one can be increasingly sure that the models are related eventually to a point where the distance between two models is a clear outlier in the distribution of distance from one of those models to the rest of the ensemble. However, for emergent constraints on future response to climate change (be it climate sensitivity or any other metric), there is little evidence that such combined multi-variate error metrics are of any use. If any strong emergent constraints do exist, then they are likely to be targeted measurements which exhibit a clear relationship to a given feedback in both their present day and in their future state.

A case in point is perturbed physics ensembles. The authors themselves have demonstrated in previous papers that such ensembles tend to exhibit common bias patterns, and when assessed in a multi-variate context such as a rank-histogram analysis, these ensembles appear under-dispersive in comparison to MMEs such as CMIP. However, such ensembles can also exhibit a greater range of climate sensitivity than the CMIP archive. As such, there is no single space into which the models can be projected which is informative about likely future response, and model dependency.

What is likely (and this is a failure of the existing literature, as well), is that interdependency is not a single property of a model pair, rather it is conditional on the question being asked. The set of all possible outputs from a climate model is high dimensional, and it is likely that only a subset of these outputs is relevant to a given aspect of future response. To take a trivial example: two models which share a sea-ice component would exhibit non-independent results for high latitude warming, but perhaps very little practical codependency for assessing rainforest dieback.

With this in mind, I don't see data mining of the inter-model distance structure as necessarily problematic as long as one is aware of what the distances represent (and I would be the first to admit that there is more to be done on this topic). With a carefully chosen metric, the existing literature has shown that the output data from a model can very nicely identify inter-model relationships – and I believe we would be foolish not to use these relationships directly in our analyses (rather than using them indirectly to confirm or refute prior suspicions of interdependency).

The second part of the paper then goes on to discuss how the results of a large ensemble of simple models can be used to assess the degree to which two potentially observable model outputs are conditionally dependent given knowledge of the model's climate sensitivity. Although this discussion is interesting, I would encourage the authors to either formulate it into a separate paper or to make a stronger case for how the concepts are related, as I feel it currently confuses what seems the primary issue of the paper – that how to formally treat model interdependencies in a small, structurally diverse ensemble.

In summary, the authors raise an important point that the existing literature has yet to provide a formal solution for addressing model interdependency which can be implemented into a well-defined mathematical framework. The paper identifies how model output data might be used to test dependency which might be usable in a formal sense, but in its current form does not demonstrate how this information could be practically used. I would encourage the authors to provide a clear case study of how such information would update a projection which might be made with the multi-model archive, and discuss more fully how their approach could be practical in the context of an ensemble without clearly defined prior expectations of model interdependency.

Ben Sanderson

---

## Referee Comment (RC2) · Anonymous Referee #2 · 26 Sep 2016

Review – Annan and Hargreaves (model independence)

The issues (plural) that are discussed in this paper are important, but the discussion is muddied by the consideration of independence in two (admittedly related) problems – model independence and the independence of constraints on climate system properties.   I agree with the first reviewer that it might not be helpful to attempt to discuss both in the same paper.

Model independence:

From a probabilistic perspective, independence is discussed in terms of "events", which are collections of elements (subsets of a sample space) that describe the multiple ways in which an "event" might occur.  From a statistical perspective, this translates into a discussion in terms of random variables (functions defined on the sample space) where equations such as $X = x$ or $X \in (x-\varepsilon, x+\varepsilon)$ for some small $\varepsilon > 0$ describe events. Here X indicates a random variable (a function that maps the sample space onto the space is observed), and x indicates a realization of that random variable (the particular value that is observed).

The only way I can conceive of the question of model independence is to start with the notion that we have available an ensemble of realizations $\{m_1, \dots, m_n\}$ of random variables $\{M_1, \dots, M_n\}$ where
- $m_i$ is the model (the entire model, not just a simulated temperature or whatever) that is the end-point of a model development process or the end point of an effort to set model parameters, and
- $M_i$ represents all of the different outcomes that would have been possible as a result of the i'th model development / parameter selection process.

There are constraints on $M_i$ that originate from the laws of physics (thus realizations of $M_i$ cannot simply be random collections of code), but within those constraints, one could in principle consider whether random variables $M_i$ are independent or perhaps even identically distributed.

The general question is not tractable in my view (whether you are a frequentist or a Bayesian) because while we may have a general notion of how to construct the sample space, we are unable to describe how that space is sampled by the model development / parameter selection process, and thus we are unable to describe the distribution of $M_i$. We might only suspect that these random variables are not all independent, or at least, that they are not identically distributed, since priorities, resources and stopping rules for the model development process differ between modelling centres. These efforts also sometimes lead to multiple versions by "branching" the model development process close to the time when the development process ends, presumably leading to dependence and common biases. Evidence that we see in the CMIP experiments that is suspected to be due to lack of model "independence" because it is associated with structural commonalities

between models, might actually be evidence that the $M_i$ are not identically distributed. That is, the two i's in "iid" might be confounded.

More specific questions, where something is known about the sampling process, such as in perturbed parameter experiments using latin hypercube sampling, are more tractable. In this case lack of independence presumably arises due to changes in nonlinear interactions between parameterized processes when parameter values change.

Independence of constraints:

While the Bayesian formalism used to explore this question is the same as that used by the authors to explore the question of model independence, the question is rather different in that it concerns the independence of observables. Thus conceptually, the source of randomness that imparts distributions on the observables is rather different. I agree with reviewer 1 that this would more appropriately be discussed in another paper.

Some specific comments (page number, line number):

1,16: This statement supposes that identical initial conditions are used each time – which does not reflect the way in which multi-simulation ensembles with a given model are constructed.  Typically, initial conditions are varied between runs so that different realizations of internal variability can be sampled across the ensemble.

1, 19: Why would the use of known, accepted physics be a problem?

2, 11: "magnitude" ➔ "potential magnitude".

3, 14-15: This statement implies that we know the truth – but reality as we know it is subject to a substantial amount of observational uncertainty and the effects of unforced internal variability. Whether truly independent or not, I think it is reasonable to think that internal variability is effectively beaten down to very small levels in the multi-model ensemble mean. In contrast, this is a source of uncertainty in observations that can only be reduced by increasing the length of the observational record.

4, 20: There seems to be a small unintentional double entendre here. "Value" can be interpreted in two different ways – you could debate whether users are overly fixated on the equilibrium climate sensitivity (a property of the climate system that is not directly observable), or you could be concerned about actually estimating this number.

5, 1-5: It might be worth pointing out that sequential updating does not necessarily require independence, but does require less than full dependence to be useful, and an understanding of the dependence structure.

5, 19-20: The description on lines 13-19 is not specific to discrete or continuous distributions. Rather, the way in which this translates into statements concerning discrete or continuous distributions depends upon the nature of the random variable (i.e., the function that maps points in a sample space onto the values that are observed).

6, 6-9: It seems to me that Box 12.2 was very clear – they described the state of the literature by writing "The peer-reviewed literature provides no consensus on a formal statistical method to combine different lines of evidence", and by providing reasons for that assessment in the preceding discussion. That doesn't strike me as being "strange" at all. Consensus would imply an approach that is used broadly across the community – and evidently that did not yet exist at the time when the AR5 report was written.

10, 2: This looks like a classic bias/variance trade-off.

10, 32-33: This begs the question of why this hasn't been done in this paper for CMIP5.

12, 8-9: Some justification for these priors would be appropriate.

---

## Author Comment (AC2) · 26 Sep 2016

Thank you for the comments. Here we comment on some of the points raised.

The reviewer questions the structure of the paper. We did find this manuscript challenging to organise while writing. A major motivation of this work was to investigate the similarities and differences in how the concept of independence applies to the two situations considered here, making one paper seems a natural choice. However it might be a better read if rather than jumping backwards and forwards between the two cases as in the current manuscript, we first present a full analysis of model independence, then consider independence of constraints as a shorter second part, identifying key similarities and differences in the application of the theory. Therefore we propose to reorganise the paper on this basis, at which point it might be clearer whether a full split

into two separate papers would be beneficial. Such a split could risk obscuring the link that exists between the two cases through the mathematical theory, and may also require a significant amount of repetition.

The reviewer notes that we do not make explicit recommendations regarding the treatment of dependent models. This was a deliberate decision. We don't think there is necessarily a single correct approach to take, although we do agree that this should be discussed or at least raised in the manuscript. One obvious choice would be to eliminate all dependency by simply removing dependent models from the ensemble prior to any analysis. This is at least in principle possible with the definition as presented, since independence is an absolute property rather than a matter of degree. It may also be possible to devise a down weighting according to the degree of independence. It seems likely that a low level of model dependence could reasonably be ignored without significantly affecting scientific conclusions. We certainly agree with the reviewer's point that the space of outputs is high dimensional and that dependence may be exhibited in some dimensions and not others. If the models which are (near-)replicated were themselves unremarkable and had no particular characteristics or a priori biases, then we would not expect eliminating a small number of dependent models to have any systematic effect on model projections or performance. On the other hand, model dependence would certainly be an important issue if a single model was to dominate an entire ensemble through say the inclusion of a huge initial condition ensemble. This is a situation that would probably be appropriately handled by scientists' intuition, but we think it is worthwhile to have a mathematical theory to underpin such treatment. We will discuss these issues in more detail in the paper.

The reviewer's comment in the middle of page C2 is hard to interpret. It seems to us that if we accept that "the ensemble has a common bias in its present day state" then this is indeed to conclude that it is not truth-centred.

The review only briefly discusses the section relating to constraints on climate sensitivity. An important point that we emphasise is that, as with the earlier case, the concept

of independence is again a fundamentally subjective one. The purpose of the example is to illustrate that nevertheless, independence can be investigated within a specific modelling framework, and can then be considered as an objective and identifiable fact conditional on that framework.

---

## Referee Comment (RC3) · G. Abramowitz (Referee) · 27 Sep 2016

I applaud the intent of this paper and its back-to-principles approach, as opposed to the pure pragmatism (potentially at the expense of principle) that does seem common in this field. It is also well written, to the extent that I have not offered any of the usual spelling, grammar or semantic corrections.

I'm not entirely sure how much it actually achieves, however. It borders on an opinion piece, in that it espouses a particular approach to framing dependence in climate science, without demonstrating in detail how the approach actually might work. Given there is not a great deal of work in this emerging area, this is not necessarily terminal for the paper at all, but at the same time it does not contribute a great deal new that can be of practical use, at least as it stands. I found it very hard to decide how to respond.

[Figure]

General comments:

I confess that I'm personally uncomfortable with a definition of model independence that does not require reference (via observations) to the system that is being estimated by the models. While I appreciate the formalism of starting from an absolutely unambiguous definition of statistical independence, it nevertheless seems counterintuitive to me to discuss independence of estimates of the climate system without any reference to it. There are clearly many other published attempts that do this (including my own), so I'm certainly not going to suggest this renders a paper unpublishable.

Even in the case of Abramowitz and Gupta (2008) and Sanderson et al (2015) - which I actually think are conceptually similar in their treatment of dependence - I feel that defining dependence in terms of model distances (without reference to observations), separately calculating model-observation distances, and then combining them to form weights, is not ideal.

My objection to raw inter-model distance (e.g. RMS) being the metric for independence is that we cannot a priori tell whether a given distance 'd' between two models is 'good' (suggesting they're independent) or 'bad' (suggesting they're dependent) if we have no observationally-based reference point. That is, these models are independent if 'd' is large relative to their distance from observations, and dependent if 'd' is small relative to their distance from observations. So while we can account for this in weights as the two papers above do (effectively defining conditional dependence), I feel we still do not have a 'clean' proxy measure for dependence in this context (i.e. distance in this sense does not equate to independence). This dissatisfaction is what eventually lead to the work in Bishop and Abramowitz (2013) and Abramowitz and Bishop (2015).

Next, despite the use of some examples in the paper, they are not concrete examples that illustrate how the principles espoused here can actually translate into a useful improvement in predictions, once dependence is accounted for. In fact, there seems no guidance in how to account for dependence once it is recognised. Without this, it's

difficult to get a handle on exactly what the paper is prescribing we do, beyond adhere to statistical formalism.

I also wasn't clear why the paper covered two topics that in practice seem somewhat unrelated - independence of models making projections and independence of constraints. As it stands, this confuses the focus of the paper. It might well be better to stick to the first of these (which is what the literature review speaks to) and actually illustrate the kind of solution it might provide.

Specific comments:

On line 26 of page 3, the authors suggest that "this approach has the weakness that models that agree because they are all accurate will be discounted, relative to much worse models, without any allowance being made for their good performance relative to reality." Yet, at least as I understand it, the example they give on pages 8,9 & 10 - suffers from exactly the same problem, does it not?

I also feel that the above reference (to Abramowitz and Gupta (2008)) is not quite representative. In that paper - which I now disagree with for the reason outlined above - models were down-weighted if their outputs were similar and they were not close to observations, not simply if their outputs were similar, as suggested here. Indeed the second sentence (L26) is actually misleading, as Equation 2 and the discussion around it in that paper directly address this point and accounts for this issue.

P5, L7: What about the case when the multiple lines of evidence disagree? I would argue the result is not always more precise as suggested here.

Gab Abramowitz

---

## Author Comment (AC3) · 29 Sep 2016

Thank you for the comments. As an initial reply to your discussion of model distributions, we are not really very sure how it applies to our manuscript. It was perhaps an oversimplification to imply as we did that all models could be assumed to have been sampled from some specific distribution, but this is not fundamental to our approach. The uncertainty referred to in our manuscript relates specifically to the lack of knowledge of the researcher regarding a (new) model's outputs. This does not require (at least, from our perspective) the creation of any detailed bottom-up statistical structure from which the model is deemed to have be sampled in its creation, it merely requires the recognition that the researcher's expectation of the model's outputs may be influenced by knowing the outputs of its relative or relatives - or more generally, knowledge of which models it is likely to be more or less similar to. We do not understand what

exactly is being claimed as intractable, as this comment appears to be contradicted by our example (which, though simple, does use real model data).

---

## Author Comment (AC4) · 29 Sep 2016

Thank you for the comments, to which we here provide a partial reply.

We think it is important to separate out several ideas which are, we believe, conceptually separate. Model duplication (and near-duplication) can be in our view considered separately from model performance. It is just as easy to tweak parameters or structure of a "good" model (however this is defined), and add this near-replicate to the CMIP ensemble, as a "bad" one. Thus it is not obvious to us how or why model performance could be a useful indicator in assessing model duplication. We emphasise that we certainly consider assessment of model performance to be important, especially when predictions are being made (and have published several papers relating to this). However, it is not clear that this can be usefully linked to the question of model

independence. A challenge that our manuscript presents is to discover whether and how the concept of probabilistic independence can be applied to measures that do take account of performance (presumably in addition to inter-model differences). It may be possible to do so, and this would be useful in explaining what the terms of this (conditional) independence actually are. We consider that a significant achievement of the paper is its presentation of a mathematically sound foundation for the discussion of model dependence which can be built on in future work.

We can certainly discuss in more detail the effect of model dependence on prediction, and present some analysis of this. It is amenable to a relatively straightforward theoretical analysis. For example, if we have 15 independent models (according to our definition) and then 5 of these (selected at random) are replicated once each, the effective ensemble size is reduced to around 13, just a little smaller than if the 15 independent models had been used (though more markedly smaller than the apparent ensemble size of 20). This will be expected to have a very small effect on ensemble performance, e.g. as suggested by Figure 3c,d of Knutti et al (2010) where it is shown that the ensemble mean of 20 or 21 randomly-selected models typically has a very slightly worse performance than when all 23 are used. If, on the other hand, a future CMIP ensemble contained a massive ensemble from one modelling centre alone, then this could have a much more significant effect, reducing the effective ensemble size to one or two. Again, Figure 3 c,d of Knutti et al indicates the likely effects of this on the ensemble mean, with an increase in expected RMS error of up to about 50%. In this case the ensemble spread would also collapse, leading to additional problems. Of course most researchers already limit themselves to a single ensemble member from each model when performing multi-model analyses, but the principle here applies also to near-replication through dissemination of a model to multiple modelling centres. A new paper bears this out, with Leduc et al (2016) finding very little different in mean projections when "institutional democracy" is imposed (though the differences in spread are more marked, especially when the ensemble is reduced most severely). Therefore, we expect that accounting for model dependence (as defined and demonstrated

in the manuscript) will have very little effect on predictions, but it could potentially have a larger effect in future iterations of CMIP.

Specific comments

p3 l26 - yes, this is correct. The approach presented here does not account for model performance as discussed above.

p5 l7 This is possible, but we do not expect to have contradictory evidence. The point about always expecting to learn from new evidence is mathematically derivable, and is not contradicted by there being some occasions where we "unlearn". Such events must however be expected to be relatively rare (expectation here being used in the formal probabilistic sense).

References:

Knutti et al 2010 "Challenges in Combining Projections from Multiple Climate Models" Journal of Climate

Leduc et al 2016 "Is institutional democracy a good proxy for model independence ?" Journal of Climate, in press

---

## Author Response (AR1)

Dear Professor Sun,

We have uploaded our revised manuscript. We apologise for the long delay, but we found the comments and discussions at NCAR in December particularly useful and interesting, and believe that the manuscript is substantially improved as a result.

We would like to draw your attention to two major changes to the manuscript.

1. Several reviewers questioned the structure of the paper and in particular the blending of discussion of model ensembles with constraints on the equilibrium climate sensitivity. We have now separated these two topics into two Parts, which we believe should make the paper easier to follow. We are presenting these two parts together as a single paper in order to make clear the similarities and differences between the underlying theory in both cases.

2. We now introduce a weighting scheme that can account for model dependency (Section 4.2). This again was something that several other commenters asked about and we realised after a bit of thought that it was in fact fairly straightforward to implement. The method, while very simple, does present a concrete approach to dealing with model dependence which was missing in the original manuscript.

Regards,

James Annan

Reply to Ben Sanderson

Thank you for your comments. This response is in addition to our previous comments made as part of the Interactive Discussion.

We would like firstly to draw attention to two major changes to the manuscript.

1. Several reviewers questioned the structure of the paper and in particular the blending of discussion of model ensembles with constraints on the equilibrium climate sensitivity. We have now separated these two topics into two Parts, which we believe should make the paper easier to follow. We are presenting these two parts together as a single paper in order to make clear the similarities and differences between the underlying theory in both cases.

2. We do now introduce a weighting scheme that can account for model dependency (Section 4.2). This again was something that several commenters asked about and we realised after a bit of thought that it was in fact fairly straightforward to implement.

As a consequence of point 1 in particular, the diff file is particularly unhelpful, as the latexdiff utility is unable to parse the manuscript correctly. It is really not possible to describe the changes in detail, as every section has been changed (and often completely rewritten) in order to separate out the two parts. However, the underlying argument and reasoning of the paper is unchanged.

There are the following significant additional changes:

A second demonstration of model dependence is now included in Section 4.1, which uses the discrete set of model outputs without the need for any parametric and distributional assumption such as a multivariate Gaussian distribution built around these outputs. While the latter

approach is widespread and probably not unreasonable, we think it is attractive to be able to present a demonstration which makes a minimum of additional assumptions.

The equations for the simple climate model in Section 6 were previously garbled in their presentation. However, the model itself, and the results, are unchanged.

We hope that our previous response was adequate in terms of the specific points discussed at that time. In particular, we do not expect that accounting for model dependency will change existing climate model ensembles in any systematic way, because the current level of dependency is moderate and we do not have any prior expectation that the dependent models are atypical. We now include in the manuscript an explanation along the lines of our initial reply to reviewer Gab Abramowitz. In short, we would anticipate the performance of the ensemble mean to be insignificantly changed for any moderate estimate of model dependency and duplication. Conversely, in the event of a future ensemble being dominated by a small subset of highly replicated models, then accounting for model dependence could become much more important.

Reply to Reviewer #2

Thank you for your comments.

We would like firstly to draw your attention to two major changes to the manuscript.

1. Several reviewers questioned the structure of the paper and in particular the blending of discussion of model ensembles with constraints on the equilibrium climate sensitivity. We have now separated these two topics into two Parts, which we believe should make the paper easier to follow. We are presenting these two parts together as a single paper in order to make clear the similarities and differences between the underlying theory in both cases.

2. We now introduce a weighting scheme that can account for model dependency (Section 4.2). This again was something that several other commenters asked about and we realised after a bit of thought that it was in fact fairly straightforward to implement. The method, while very simple, does present a concrete approach to dealing with model dependence which was missing in the original manuscript.

As a consequence of point 1 in particular, the diff file is particularly unhelpful, as the latexdiff utility is unable to parse the manuscript correctly. It is really not possible to describe the changes in detail, as every section has been changed (and often completely rewritten) in order to separate out the two parts. However, the underlying argument and reasoning of the paper is unchanged.

In reply to your specific points (and in addition to our earlier reply of 29th September in the Interactive Discussion phase):

1,16: True but not really important for our argument. Even with IC ensembles the long-term climate change will

be very similar. We have changed the wording here slightly.

1, 19: We don't think this would in principle be a problem, on the other hand, it may be difficult to distinguish this situation from one in which a ubiquitous error exists, especially when this error is embedded in a complex system with multiple compensating errors.

2, 11: accepted

3, 14-15: It is quite clear that the error of the ensemble mean is substantially larger than any realistic estimate of observational uncertainty or internal variability, at least for many of the more robustly observed climate variables. Of course we agree that for some poorly-observed variables, it might not yet be possible to demonstrate this.

4, 20: A good point and it's been changed from "value" to "magnitude".

5, 1-5: substantially reworded.

5, 19-20: Ok, but we think that the way we've presented it should be clear to most climate scientists.

6, 6-9: Deleted

10, 2: Possibly, but we are not clear how that concept applies to our example.

10, 32-33: CMIP5 is presented as a future test case.

12, 8-9: The priors are just subjective choices chosen to give reasonably acceptable results. The model does not have to relate well to the real world in order for the point to be made, however.

There are also the following significant additional changes:

A second demonstration of model dependence is now included in Section 4.1, which uses the discrete set of model outputs without the need for any parametric and distributional assumption such as a multivariate Gaussian distribution built around these outputs. While the latter approach is widespread and probably not unreasonable, we think it is attractive to be able to present a demonstration which makes a minimum of additional assumptions.

The equations for the simple climate model in Section 6 were previously garbled in their presentation. However, the model itself, and the results, are unchanged.

Reply to Gab Abramowitz

Thank you for your comments. This response is in addition to the reply we gave during the Interactive Discussion phase.

We would like firstly to draw attention to two major changes to the manuscript.

1. Several reviewers questioned the structure of the paper and in particular the blending of discussion of model ensembles with constraints on the equilibrium climate sensitivity. We have now separated these two topics into two Parts, which we believe should make the paper easier to follow. We are presenting these two parts together as a single paper in order to make clear the similarities and differences between the underlying theory in both cases.

2. We now introduce a weighting scheme that can account for model dependency (Section 4.2). This again was something that several other commenters asked about and we realised after a bit of thought that it was in fact fairly straightforward to implement. The method, while very simple, does present a concrete approach to dealing with model dependence which was missing in the original manuscript. To reiterate what we said in our earlier reply to you, however, accounting for model dependence in this way should not be expected to affect ensemble performance to any significant extent, not least because model (near-)replication and hence dependence is essentially unrelated to model performance. The change in effective ensemble size by down weighting related models is very small. However, in a hypothetical future iteration of CMIP which could contain a large number of replicates of a few models, this could in principle become a much more important matter.

As a consequence of point 1 in particular, the diff file is particularly unhelpful, as the latexdiff utility is unable to parse the manuscript correctly. It is really

not possible to describe the changes in detail, as every section has been changed (and often completely rewritten) in order to separate out the two parts. However, the underlying argument and reasoning of the paper is unchanged.

There are the following significant additional changes:

A second demonstration of model dependence is now included in Section 4.1, which uses the discrete set of model outputs without the need for any parametric and distributional assumption such as a multivariate Gaussian distribution built around these outputs. While the latter approach is widespread and probably not unreasonable, we think it is attractive to be able to present a demonstration which makes a minimum of additional assumptions.

The equations for the simple climate model in Section 6 were previously garbled in their presentation. However, the model itself, and the results, are unchanged.

We have extended our discussion of previous work to include both Abramowitz and Gupta (2008) and Bishop and Abramowitz (2013). We had not previously realised how substantial the differences were between these two approaches.

We also refer the reviewer to our previous reply of the 29th September 2016, in the Interactive Discussion.